# A Rare Case of Orbital Castleman Disease with Overlapping IgG4-Related Disease

**DOI:** 10.3390/medicina59081381

**Published:** 2023-07-28

**Authors:** Li-Ching Liu, Yann-Guang Chen, Nien-Tzu Liu, Yi-Hao Chen, Ke-Hung Chien

**Affiliations:** 1Department of Ophthalmology, Tri-Service General Hospital, National Defense Medical Center, Taipei 11490, Taiwan; ophelia30330@gmail.com (L.-C.L.); rainmaker1022@gmail.com (Y.-G.C.); doc30879@mail.ndmctsgh.edu.tw (Y.-H.C.); 2Department of Pathology, Tri-Service General Hospital, National Defense Medical Center, Taipei 11490, Taiwan; s1326amy@yahoo.com.tw

**Keywords:** multicentric Castleman disease, IgG4-related disease, orbital tumor, histopathology

## Abstract

Multicentric Castleman disease (MCD) is a systemic lymphoproliferative disorder that can lead to mass lesions in various body parts, including the lungs, kidneys, and extranodal sites. Meanwhile, orbital Castleman disease is extremely rare. Immunoglobulin G4-related disease (IgG4-RD) is a recently recognized fibroinflammatory disorder and is characterized by the formation of tumor-like lesions with lymphoplasmacytic infiltrates, which are enriched in IgG4-positive plasma cells and may present with a characteristic storiform pattern of fibrosis to variable degrees. In this study, we report a case of a 67-year-old Taiwanese man with a 7-year history of bilateral eyelid swelling and proptosis. Orbital magnetic resonance imaging revealed soft tissue lesions in the bilateral intraconal region, demonstrating strong enhancement in the lacrimal glands, and extension into the bilateral infraorbital foramen, suggesting an orbital lymphoproliferative disease. The histopathological results of the intraorbital tumor excision were suggestive of a plasma-cell-predominant mixed-cell variant of MCD. However, the patient also showed definitive signs of IgG4-RD, including lacrimal gland enlargement and histopathological results of plasmacytosis, fibrosis, and germinal centers, with an increased ratio of IgG4 cells and elevated serum IgG4 levels. This case suggests a potential interacting pathway between these two disease entities that needs further studies.

## 1. Introduction

Orbital Castleman disease is a rare extranodal occurrence of a benign hyperlymphoproliferative disorder that primarily affects the orbital tissues and was first described by Gittinger et al. in 1989 [1]. The disease is characterized by abnormal proliferation of lymphoid cells within the orbit, leading to the formation of one or more masses or tumors. As a result, patients may experience symptoms such proptosis, diplopia, pain, and visual disturbances [2]. Clinically, Castleman disease can be divided into unicentric Castleman’s disease and multicentric Castleman disease (MCD). MCD is further subdivided into HHV-8-positive MCD and HHV-8-negative/idiopathic MCD (iMCD). Histopathologically, Castleman disease can be classified into three variants: hyaline-vascular, plasma cell, and mixed cell types [3]. There is another fibroinflammatory disorder called IgG4-related disease (IgG4-RD), which has gained recognition in the medical community recently. It is characterized by the formation of tumor-like lesions with lymphoplasmacytic infiltrates, enriched in IgG4-positive plasma cells, and might present with characteristic storiform pattern fibrosis in variable degrees [4,5,6]. The clinical features of IgG4-RD and iMCD exhibit similarities, leading to challenges in differentiating between these conditions [7]. In this study, we report a case of orbital Castleman disease with clinically overlapping IgG4-RD. This study was approved by the Institutional Review Board of the Tri-Service General Hospital in Taiwan.

## 2. Case Report

A 67-year-old man presented with bilateral eyelid swelling and proptosis for 7 years and had medical histories of allergic rhinitis and gout. Initially, a painless, progressively enlarging tumor was present in the left upper eyelid. The pathological findings from excisional biopsies in 2013 and 2019 yielded reactive lymphoid hyperplasia.

In 2020, a year later, the patient reported ocular irritation, blurred vision, progressive swelling of the right upper eyelid, and proptosis of the right eye without fever, night sweats, and weight loss. The best-corrected visual acuity was 20/25 and 20/32 in the right and left eye, respectively. Hypotropia and head tilt to the right with mild limitation in the abduction of the right eye were noticed (Figure 1). A painless, well-defined, non-movable palpable soft mass was noted on the right upper eyelid. Serology tests revealed an elevated erythrocyte sedimentation rate and IgG-4 level. Levels of serum kappa and lambda-free light chains were also increased. HHV-8, Epstein–Barr virus, and Human Immunodeficiency Virus testing were negative. Magnetic resonance imaging (MRI) revealed soft tissue lesions involving bilateral lacrimal glands, suggesting an orbital lymphoproliferative disease (Figure 2).

Proptosis and swelling of the right upper eyelid, along with limitation of lateral gaze in the right eye, were observed before the operation. The Hertel exopthalmometric values were 26 mm for the right eye and 21 mm for the left eye, with an inter-orbital distance of 113 mm.

Excisional biopsy with lateral orbitotomy and primary reconstruction of the right eye was performed. The specimen consisted of four small pieces of lacrimal gland tissue measuring up to 3.6 × 2 × 1.8 cm in size. In addition, they were gray in color and soft in consistency. Immunohistochemical stains confirmed the presence of atypical lymphoid hyperplasia of the lacrimal gland tissue with features consistent with those of the plasma-cell-predominant mixed-cell variant of idiopathic MCD (iMCD) (Figure 3A–E). However, the patient also showed definitive signs of IgG4-RD, including an atopic history, elevated serum IgG4 level, imaging that revealed bilateral lacrimal gland enlargement, and histopathological results of IgG4 + cells/IgG + cells > 40% (Figure 3F).

Outpatient follow-up was arranged after surgery, and the orbital condition remained stable without signs of recurrence of the right eye in a month’s follow-up based on clinical examination. The patient was referred to the hematology and oncology department for systemic evaluation and suggestion of possible steroid therapy. However, the patient did not attend the follow-up due to personal reasons.

## 3. Discussion

To differentiate between orbital iMCD and IgG4-RD is a clinical challenge due to its rarity. We present a rare case of pathological-proofed orbital iMCD with clinically overlapping characteristics of IgG4-RD.

MCD is a systemic lymphoproliferative disorder that can lead to mass lesion formation in various parts of the body, including the lungs, kidneys, and extranodal sites. Although orbital tissue involvement is relatively rare due to the lack of lymphatic components, previous cases of MCD affecting the lacrimal gland have been reported [8,9]. Venizelos et al. reported that orbital involvement might be the initial clinical presentation of CD and those with unicentric hyaline-vascular CD cases showed no association with systemic symptoms and were rarely recurrent [10]. In our case, the patient mainly presented with localized disease that had histopathological findings consistent with the findings of iMCD. There were no systemic symptoms such as B symptoms, anemia, or hepatomegaly observed.

IgG4-RD is a newly recognized immune-mediated disease that can result in the formation of tumors and can affect multiple systems, causing diagnostic challenges [4]. The diagnosis of IgG4-RD relies on clinical manifestations, such as atopic history, exocrine gland involvement, and elevated serum and tissue IgG4/IgG ratios. However, an elevated IgG4/IgG ratio in the serum or tissue is not a specific diagnostic marker for IgG4-RD. A confirmatory biopsy with complete immunohistochemistry is needed to differentiate IgG4-RD from malignancy or other IgG4-RD mimics [11]. In ocular IgG4-RD, lacrimal glands are primarily affected bilaterally [12]. In our case, definitive signs characteristic of IgG4-RD were present. These included imaging that showed bilateral lacrimal gland involvement, pathological results of plasmacytosis, fibrosis, and an IgG 4/IgG ratio > 40%.

Both patients with MCD and IgG4-RD present with extranodal involvement and elevated serum IgG4 levels. Generally, patients with IgG4-RD tend to be older than those with iMCD [13]. Patients with iMCD may present with fever and high C-reactive protein, interleukin (IL)-6, and IgA levels that are absent in patients with IgG4-RD. Histologically, plasmacytosis is observed in the affected tissue in both conditions, while plasma cells arranged in a sheet-like pattern are often seen in iMCD [14]. However, distinguishing between IgG4-RD and iMCD is not always possible, even if certain mentioned characteristics are fulfilled. To date, the pathogenesis of these two diseases has been found to be different, leading to distinctive treatment approaches. In IgG4-RD, IL-4 is the critical cytokine related to type 2 inflammatory reactions that often contributes to allergic reactions or atopic manifestations, whereas increased serum IL-6 levels are observed in almost all patients with iMCD [7,15]. Previous studies showed the role of helper (Th) 2 cells and T follicular helper (Tfh) cells in the regulation of IL-4 expression [16,17]. Diehl et al. concluded that IL-6 promotes Th2 differentiation and is dependent on endogenous IL-4 [18]. A recent study also suggested that IL-4 and IL-6 cooperate to induce the pro-tumor polarization of primary human macrophages. [19] Mochizuki et al. reported a case in which overlapping features of IgG4-RD and MCD were present in a single patient, which is similar to our case [20]. The possibility of cytokine activation pathway interaction in extranodal inflammatory disease may need further studies to support the overlapping clinical manifestations and similar pathological findings. The lack of universal standard differential criteria between IgG4-RD and plasma-cell-predominant MCD makes diagnosis challenging in clinical practice. A comprehensive evaluation, including immunohistopathological characteristics, laboratory tests, and clinical treatment response, should be conducted to guide appropriate treatment decisions and assess prognosis.

## 4. Conclusions

Both MCD and IgG4-RD are tumor-forming diseases that can affect orbital tissue. Similar histopathology findings and overlapping clinical presentations may suggest the possibility of shared pathogenesis or interacting pathway behind these two conditions.

## Figures and Tables

**Figure 1 medicina-59-01381-f001:**
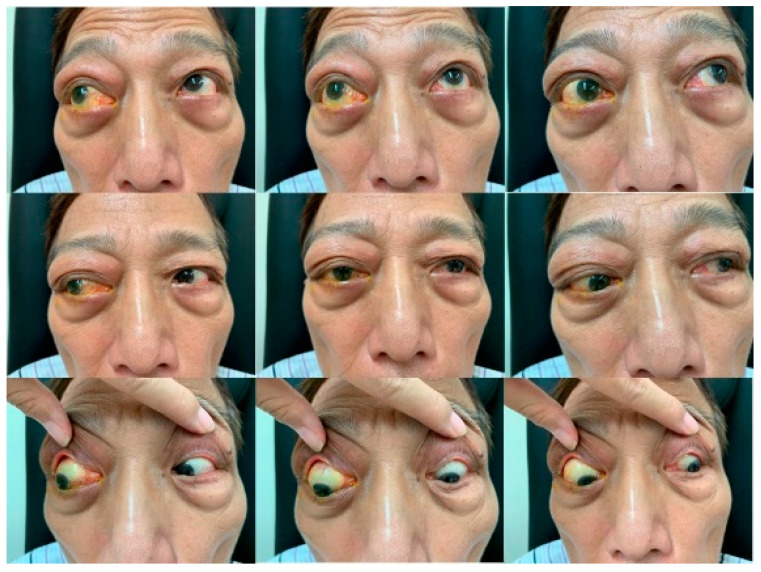
Composite 9-gaze photograph.

**Figure 2 medicina-59-01381-f002:**
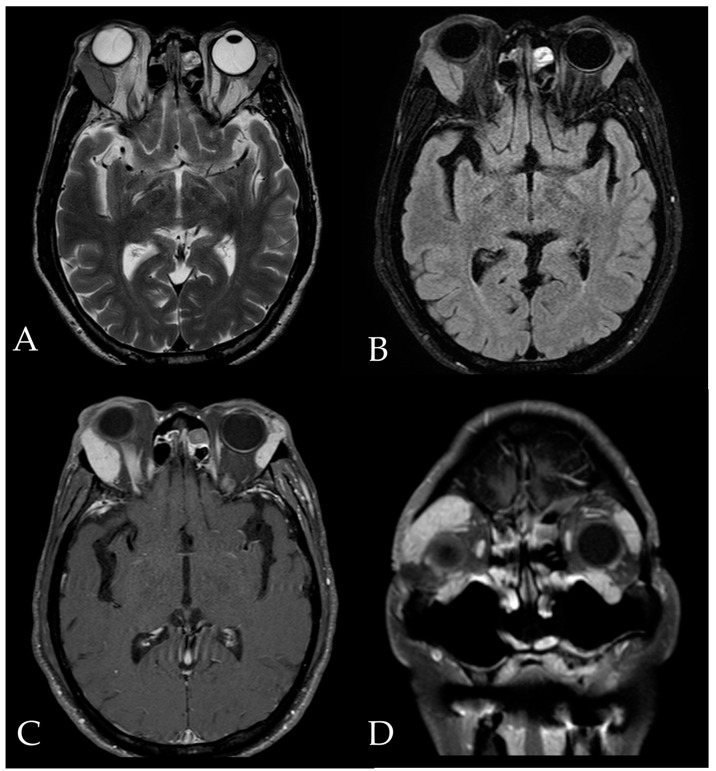
Magnetic resonance imaging (MRI) findings of orbits. T2-weighted (**A**) and T2 fat-suppressed (**B**) MRI images revealed the presence of soft tissue lesions in the bilateral orbital cavity with lacrimal gland involvement, and extension into the intraconal region and the bilateral infraorbital foramen. T1-weighted contrast-enhanced images in axial view (**C**) and coronary view (**D**) showed homogeneous enhancement of the lesions, suggestive of orbital lymphoproliferative disease.

**Figure 3 medicina-59-01381-f003:**
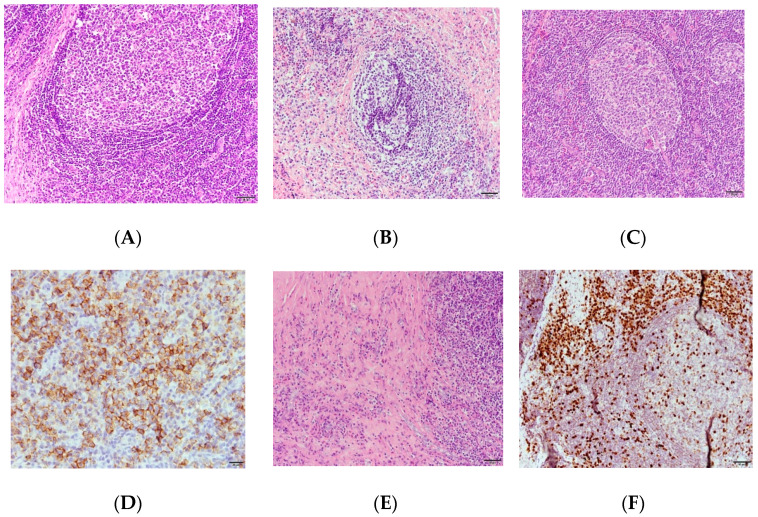
Histopathology showing typical characteristics of Castleman disease. Histopathology revealed many hyperplastic and few regressed germinal centers with concentric mantle zones (**A**). A “twining” pattern (**B**), characterized by two germinal centers within one follicle, was also observed. The presence of occasional follicular dendritic cells and radially penetrating vessels revealed a lollipop-like appearance (**C**). Distinct interfollicular polyclonal plasmacytosis (CD-138) with lambda light chain predominance (**D**) was observed. Focal fibrosis with a slight increase in eosinophils (**E**) was also noted. These findings were consistent with those of the plasma-cell-predominant, mixed-cell variant of iMCD. Immunohistochemical staining demonstrated a ratio of IgG4 + cells/IgG + cells above 40% (**F**).

## Data Availability

All data are included in the manuscript.

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
