# Peer review of "A Rare Case of Orbital Castleman Disease with Overlapping IgG4-Related Disease"

_medicina, 2023, doi:10.3390/medicina59081381_

Round 1

Reviewer 1 Report

This is an interesting case of a patient presenting with features of 2 lymphoproliferative diseases. The presentation of disease and pathological features make it difficult to differentiate between the two in some patients. This case, as the authors rightfully note, might suggest a link between the diseases.

A few minor points:

·      Completeness: The presentation is lacking in terms of workup and treatment.
Both IgG4-RD and MCD can cause multifocal disease. Was PET-CT or total body imaging carried?
Also, both diseases are usually treated with steroids, biologics or anti-inflammatory drugs. Did the patient received any further treatment?
Details about the surgery are also missing (Was it excisional or incisional? Debulking or complete removal? What happened to the contralateral gland?).
How long was the follow-up following surgery? How was the patient followed-up (imaging? Lab? Clinical exam only?)

·      Robustness: Line 109 – 110, line 152: While this is an interesting idea, that can and should be discussed in the discussion (As the authors did in lines 135-139), the authors did not show any concrete data suggesting a common pathological origin or interacting pathway between the diseases, therefore in my opinion, one should be more careful with it.

·      The authors rightfully point to IL-4 and IL-6 as contributing factors to IgG3-RD and MCD, and they rightfully suggest the different attributes of these interleukins. Do they have an hypothesis regarding a common cause for the diseases?

From a style perspective, while being easily understandable, the overall presentation could be made better by careful editing. A few examples are:

·      Line 37: “In addition to …” should be re-phrased.

·      Medical history should follow the presentation (and not be revealed in the 2nd paragraph; line 52).

·      Line 103 should be moved to an appropriate location.

·      Line 107- 108 – needs editing

·      There are more minor glitches along the article, careful editing is suggested.

Reviewer 2 Report

Liu and colleagues highlight an interesting case of orbital Castleman Disease with features of IgG4 disease, suggesting a potential shared mechanism underlying these diseases. The case is interesting but can be improved through the following points:

Major:

-       Figure 2 MRI shows a T2 weighted MRI showing hypointensity of the lacrimal glands and enlargement of the right>left glands. The image does not seem to be non-fat-suppressed and there is no contrast within imaging. The paper would strongly benefit from including additional MRI figures, including a T1 post-contrast and T2 fat-suppressed images within Figure 2 and labeling images as appropriate.

-       One of the most comprehensive studies of orbital Castleman Disease is by Venizelos et al. in Survey of Ophthalmology. Recommend authors read and expand their discussion of the topic either in the Introduction or Discussion sections using the citation: Venizelos I, Papathomas TG, Papathanasiou M, Cheva A, Garypidou V, Coupland S. Orbital involvement in Castleman disease. Surv Ophthalmol. 2010;55(3):247-255. doi:10.1016/j.survophthal.2009.09.003

Minor:

-       Line 60: Can be rephrased as “HHV-8, Epstein-Barr virus, and Human Immunodeficiency Virus testing was negative.”

-       Line 79: Tumor excision should be expanded. What technique? Any intraoperative findings?

-       Figure 3. There is a Euro sign in the figure legend, which presumably reflects eosinophils, but is not labeled in the image. Recommend removing it or placing it as a label in the figure.

-       Lines 103-104: IRB approval can be placed at the end of the introduction or beginning of the case report

Round 2

Reviewer 2 Report

The authors have addended the paper to include additional discussion and references. The addition of images has significantly improved the paper.